# Out-of-equilibrium processes in crystallization of organic-inorganic perovskites during spin coating

Shambhavi Pratap [1,2], Finn Babbe [3], Nicola S. Barchi[4,5], Zhenghao Yuan [4,6], Tina Luong[4], Zach Haber[4], Tze-Bin Song [4], Jonathan L. Slack[2], Camelia V. Stan [2,7], Nobumichi Tamura [2], Carolin M. Sutter-Fella [4,8 ✉] & Peter Müller-Buschbaum [1,9 ✉]

Complex phenomena are prevalent during the formation of materials, which affect their processing-structure-function relationships. Thin films of methylammonium lead iodide ($CH_3NH_3PbI_3$, MAPI) are processed by spin coating, antisolvent drop, and annealing of colloidal precursors. The structure and properties of transient and stable phases formed during the process are reported, and the mechanistic insights of the underlying transitions are revealed by combining in situ data from grazing-incidence wide-angle X-ray scattering and photoluminescence spectroscopy. Here, we report the detailed insights on the embryonic stages of organic-inorganic perovskite formation. The physicochemical evolution during the conversion proceeds in four steps: i) An instant nucleation of polydisperse MAPI nano-crystals on antisolvent drop, ii) the instantaneous partial conversion of metastable nano-crystals into orthorhombic solvent-complex by cluster coalescence, iii) the thermal decomposition (dissolution) of the stable solvent-complex into plumboiodide fragments upon evaporation of solvent from the complex and iv) the formation (recrystallization) of cubic MAPI crystals in thin film.

[1] Physik-Department, Lehrstuhl für Funktionelle Materialien, Technische Universität München, Garching, Germany. [2] Advanced Light Source, Lawrence Berkeley National Laboratory, Berkeley, USA. [3] Chemical Sciences Division, Joint Center for Artificial Photosynthesis, Lawrence Berkeley National Laboratory, Berkeley, USA. [4] Chemical Sciences Division, Lawrence Berkeley National Laboratory, Berkeley, USA. [5] Laboratoire des Matériaux Semiconducteurs, École Polytechnique Fédérale de Lausanne, Lausanne, Switzerland. [6] Department of Chemistry, The Pennsylvania State University, University Park, PA, USA. [7] NIF & Photon Science, Lawrence Livermore National Laboratory, Livermore, USA. [8] Molecular Foundry, Lawrence Berkeley National Laboratory, Berkeley, USA. [9] Heinz Maier-Leibnitz-Zentrum, Technische Universität München, Garching, Germany. ✉email: csutterfella@lbl.gov; muellerb@ph.tum.de

Metal halide-based hybrid perovskite materials have attracted significant research and development interest due to their truly impressive and broad applicability as functional materials[1]. The techno-economic advantages[2] of hybrid perovskites, in addition to their stellar optoelectronic properties[3], arise from their facile and low-cost solution processability[4]. Spin coating is a well-established and widely utilized method for the formation of high-quality perovskite thin films. In recent years, improvements over conventionally spin-coated thin film morphologies were achieved by exploiting treatment methods such as Lewis acid-base precursor adduct engineering[5], complex intermediate driven crystallization[6], additive usage[7], intramolecular exchange[8], and antisolvent[9] driven film formation. Post deposition film treatments such as thermal[10] and solvent-vapor annealing[11] have also been explored to eliminate undesirable[12] structural constraints and are also known to lead to advantageous morphological effects such as the emergence of hierarchical microstructures within thin films[13]. Significant efforts toward controlling perovskite thin-film qualities have been undertaken because of the correlation[14] between device performance metrics and thin-film structural characteristics. Moreover, a growing appreciation of the degree to which the structural quality of thin films is determined during the initial kinetic processing of colloidal precursors has instigated a close study of the evolving processing-structure-property relationships within thin films.

Synchrotron-based in situ X-ray[15] methods have aided in elucidating the structural transformations occurring during the processing of hybrid perovskites, owing to high scattering length densities[15] of heavy lead and halide-containing molecules. While X-ray based characterizations are well suited for kinetic structure analysis, they provide limited insights on the functional response of the materials themselves. By utilizing the characteristically strong photoresponse of the materials, in situ optical methods[16,17] provide insight into the evolving optoelectronic properties of crystallizing perovskites. The interdependence of structure sizes and their optical response especially helps in understanding the evolving nature[18] of short-lived intermediates and the transformation kinetics[19,20] between material phases. Important considerations that have emerged from the above studies include the identification of the complex sol-gel[21] structures involved in the transformation, intermediate solvent-complex phases[22] involved during the assembly crystallization process[23], the kinetics of their transformation to other phases[24–27], the impact of varying the time of antisolvent dispensing[28], the importance of thermal annealing processes[27,29,30], and the sensitivity toward environmental conditions on the structure of processed thin films. Correlation of material structure and properties is usually established post-fabrication and ex situ, where measured material properties can be strongly affected by differences between in situ and ex situ environments and by the impact of other synergistic functional materials involved. Structural and physical attributes attained during material formation highly influence subsequent material properties and provide information for further ex situ investigations. The sensitivity of out-of-equilibrium physicochemical structures to the multi-dimensional space of available experimental conditions[31] makes the conclusive correlation of experimental observations to their phenomenological origins complex and time-intensive, and sometimes only accessible for observation with the development of new instrumentation.

In this work, by combining the complementary and reinforcing nature of information divulged by synchrotron radiation-based X-ray and optical metrologies within controllable in situ processing environments, we unite processing-structure-function relationships. Herein the optoelectronic response is measured by means of photoluminescence (PL) spectroscopy tracking the varying functional optical response of changing structural entities traced by means of grazing-incidence wide-angle X-ray scattering (GIWAXS), during material processing by spin coating, antisolvent drop, and subsequent annealing to unveil previously empirically inaccessible mechanistic insights of complex colloidal crystallization.

Spin-coating is a solution-based processing method, which produces non-equilibrium thin films. The crystallization of colloidal precursors of hybrid perovskites has an inherent multivariate nature and is known to lead to reproducibility issues of the film characteristics. Routes of crystal growth have been discussed[32], depicting how starting from generic precursor molecules, materials crystallize to their bulk form through multiple reaction pathways. Complementary and multimodal metrological techniques help to elucidate complex transformation mechanisms responsible for reproducibility issues. For instance, it is possible for the precursor to follow physicochemical growth pathways such as spinodal decomposition[33], which deviate from the reaction pathways involving traditionally nucleated species. Rather, for instance, chemical reactions may proceed through the formation of intermediates, which convert to their final structural form on further treatment. Conceptualizing an understanding of complex growth processes requires the fixing of processing parameters, which we have done in the present study. In this work, we actuate the advantages of real-time investigation of the evolving structure and optoelectronic properties by combining GIWAXS and PL spectroscopy while emulating the one-step antisolvent-assisted[9] crystallization of a perovskite thin film. We chose to investigate methylammonium lead iodide ($CH_3NH_3PbI_3$, MAPI), whose structural intermediates and transformation kinetics have been extensively investigated. This was done as a manner of demonstrating that much remains to be learned about out-of-equilibrium assembly processes of exemplary model systems. We report on the characteristics of the metastable structure formed by the first-order phase transition occurring during antisolvent-induced nucleation from the colloidal precursor sol. Further, there is a partial transformation of the metastable nuclei by concatenation of nanostructures leading to the formation of a solvent complex, with the solvent complex being stable against thermally induced degradation up to 100 °C. Around 100 °C, a second-order transformation process of the solvent complex to MAPI is initiated, by evaporation of the solvent from the thin film. Physicochemical reaction gradients are emergent and are attributed to differential rates of removal of strongly coordinated solvent molecules via advection of the evaporating solvent molecules from the film thickness. This mechanism of solvent removal results in a process of dissolution-recrystallization to lead to the final MAPI thin film. Further annealing leads to a ripening process of the crystalline film. Evolutionary data signatures, physical concepts, and characteristics within structure-function correlations learned from a model system of MAPI are transferable to other chemical compositions of hybrid perovskites. These insights are enabled through the development of a novel analytical cell[34], which allows complete remote control over the spin coating process, antisolvent drop, PL excitation, and the annealing protocols. The processing and measurement environments are housed within an inert gas-purged cell to curb unfavorable degradation by atmospheric oxygen and moisture [35].

## Results

A simultaneous overview of the evolution of structural and optoelectronic phases recorded by GIWAXS and PL is presented in Fig. 1. Key structural phase transitions witnessed in diffraction are presented as individual 2D diffraction images in Fig. 2.

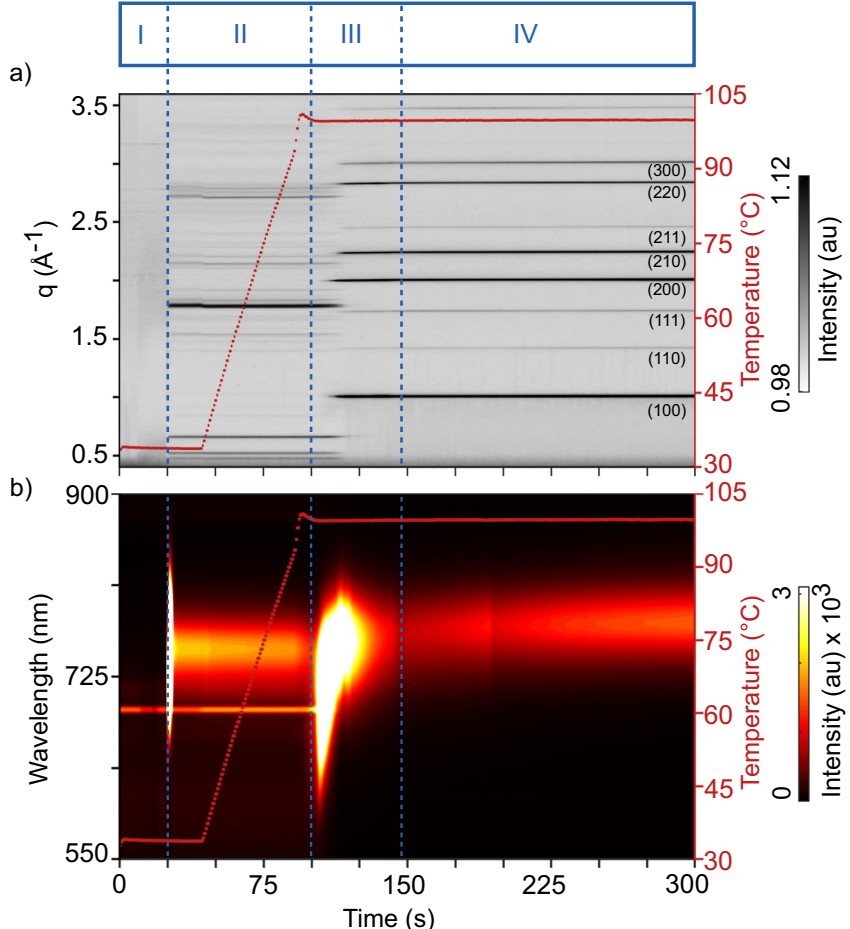

**Fig. 1 Mapping with four phases seen in GIWAXS and PL data.** Time evolution representing the four phases of the colloidal conversion process of PbI$_2$.CH$_3$NH$_3$I.DMSO.DMF precursor to a final crystalline MAPI thin film as indicated by vertical dotted lines. **a** Radially integrated GIWAXS data as a function of q position and (**b**) PL data as a function of wavelength together with the substrate temperature (right y-axis). The narrow line emission at 690 nm is related to the diffuse reflection of the laser used in the position alignment system of the beamline. Phase I shows the spin coating followed by the antisolvent drop at $t = 25$ s. In phase II, we observe diffraction from Pb$_3$I$_8$.2(CH$_3$)$_2$SO.2CH$_3$NH$_3$ (MAPI·DMSO) solvent-complex phase, which is converted into MAPI crystals in phase III during annealing.

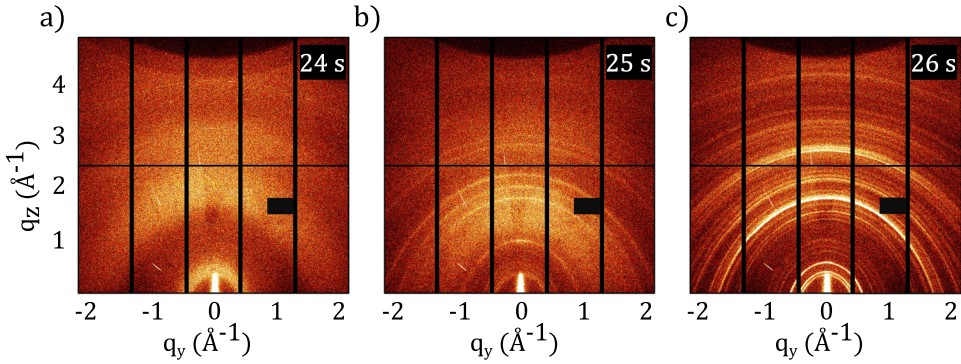

**Fig. 2 Structural conversion seen in 2D GIWAXS data.** 2D GIWAXS data as a function of the scattering vector components q$_y$ and q$_z$ during structural conversion of MAPI thin film (**a**) before, (**b**) during and (**c**) after the antisolvent drop at $t = 25$ s. Diffraction from (**a**) colloidal mixture of solid precursors and solvent molecules, (b) instantly nucleated crystalline MAPI nanocrystals and nutrient colloidal phase in the slight background, and (c) crystalline orthorhombic MAPI·DMSO solvent-complex.

During the experiment, four equilibrium phases and three transient conversion processes are registered as explained below.

**Phase I**. In the first phase (phase I, $t = 0$–24 s), the liquid precursor is spin-coated and reflected as diffuse halos centered

around 0.5 Å$^{-1}$, 1.8 Å$^{-1}$, and 3.0 Å$^{-1}$ (Fig. 2a) in the diffraction data. In line with other reports[21,22,36], this signal is attributed to scattering from the mixture of the solvent and solute phases, composed of a colloidal dispersion of chemically and structurally diverse plumboiodide scattering species[37,38]. The low scattering contrast of the halos signifies the well-intermixed state of the

solutes and solvents and the lack of any Bragg diffraction confirms the absence of long-range order within the precursor state. Phase I does not show any PL response (Fig. 1b).

Following 24 s of spin coating, an antisolvent stream is dynamically dispensed (the film is in spinning motion), resulting in rapid structural changes within the precursor phase. The diffuse halos from the precursor phase convert into low albeit distinct intensities with Bragg peaks located at $q = 1.01$, $1.78$, $2.00$, $2.24$, $2.84$, and $3.01\,\text{Å}^{-1}$ (Fig. 2b). The amorphous background and low intensities of the Bragg peaks suggest an incomplete conversion of the colloidal precursor to the proceeding state. The film is found to be isostructural in terms of peak positions with cubic MAPI. The texture of the nucleating structure is attributed to the directionality of the antisolvent dispensing, which was set normal to the substrate plane. The diffraction signal from the nucleating phase quickly transforms to the proceeding state (phase II) (Fig. 2c) suggesting the metastable nature of the causal nucleating structure (Fig. 2b).

**Phase II**. Starting at 26 s, phase II is initiated and the weak diffraction intensities from the metastable nucleated phase are converted into stronger diffraction intensities at $0.46\,\text{Å}^{-1}$, $0.51\,\text{Å}^{-1}$, $0.65\,\text{Å}^{-1}$, $1.75\,\text{Å}^{-1}$ together with several subsidiary peaks with lower intensities (Fig. 1a and Fig. 2c). The diffraction peaks correspond to the orthogonal crystalline solvent-complex, $Pb_3I_8 \cdot 2(CH_3)_2SO \cdot 2CH_3NH_3$ (MAPI·DMSO)[39–41]. In agreement with other reports[42,43], no DMF-based solvent complexes are observed in spite of the precursor solvent being DMF-rich, due to the stronger Lewis acid-base complexing ability of DMSO[5,44]. No uncomplexed $PbI_2$ is observed either. At the end of the spin coating process, at 44 s, the annealing of the thin film is initiated with a linear temperature ramp of 1 °C/s to convert the solvent complex to the crystalline perovskite. During the rest of the annealing process, the peaks from the solvent-complex of phase II remain unchanged in terms of peak positions, intensities and widths, until a temperature of 100 °C is reached and stabilized at ~100 s.

The antisolvent drop ($t = 25$ s) triggers the immediate emergence of an intense and broad PL peak (Fig. 1b) centered at around 730 nm (1.70 eV). It is suggested that the PL emission arises from the instantly formed MAPI nanocrystals, (Fig. 2b) with a polydisperse size distribution of luminescing moieties responsible for the broad FWHM of the PL emission. The peak position, which in first approximation represents the bandgaps[45,46], indicates quantum confinement of the charge carriers as typically observed in nanoparticles with a size range within ten nanometers[47–50] (expected room temperature bandgap of bulk MAPI is ~1.60 eV[51,52]).

Within the next second, the peak position shifts towards 750 nm (1.65 eV) (Figure S1a) and remains constant afterward in phase II (Fig. 3d). The redshift is due to a growth in the size of nanocrystallites, leading to a reduction in the extent of quantum confinement[50,53]. The bulk bandgap is not reached, signifying an arrested growth mechanism after a few seconds, due to reaction and diffusion-limited constraints[54,55] of the solvent-complex structures, which require thermal annealing to complete the solvent-evaporation and the transformation of the solvent-complex to crystalline perovskite. The FWHM of the PL peak at 750 nm shows significant narrowing from 130 meV ($t = 26$ s) to 110 meV ($t = 29$ s) (Figure S1a), corroborating the homogenization in size distribution. This occurs through an increase in the average sizes of structures formed by cluster coalescence of the nanoparticles, which have high correlated surface and interfacial energies. The high surface and interfacial energies are possibly strong driving forces for the cluster coalescence of the

nucleated species due to their high surface-to-volume ratio. Coalescence results in size homogenization of the particles, which results in a narrowing of the PL spectra. After the polydisperse nucleation process, the MAPI nuclei above a critical radius are expected to remain stable against cluster coalescence. These MAPI crystals, which are not bound into the solvent complex phase are hypothesized to contribute to the remaining luminescence intensity in phase II (the solvent complex by itself likely does not luminesce). Thereafter, in phase II, the PL response remained unaffected in terms of peak shape, position, and intensity, suggesting remnants of stable perovskite crystals formed during the nucleation process.

**Phase III**. On reaching 100 °C, around $t = 90$ s, the solvent complex undergoes further structural transitions caused by thermal disassociation and subsequent solvent evaporation, marking the initiation of phase III of the crystallization process. Around 103 s, Bragg peaks from a crystallizing perovskite phase start quickly gaining intensity, reaching the maximum at 120 s (Fig. 3a). The evolution of the perovskite phase (black curve) is compared against structural changes of the solvent-complex (green curve) by tracking the intensities of their Bragg peak $q_{100} = 1.00\,\text{Å}^{-1}$ and $q_{150} = 1.78\,\text{Å}^{-1}$ (Fig. 3a), respectively. The lattice spacings of the perovskite crystals increase while the peak widths narrow, for $t = 104$–120 s (Fig. 3b). Simultaneously, the solvent-complex Bragg peaks diminish in intensity. This relative intensity change between 104–120 s is attributed to the conversion of the solvent complex to the perovskite state at the film-air interface where the rate of evaporation of the solvent is expected to be the highest.

Beyond 120 s, there is a second, slow decrease in the diffraction intensities of the solvent-complex (Fig. 3a) up to 144 s, which is attributed to the removal of the solvent complex from the deeper parts of the thin film, which require longer annealing times for complete solvent removal. Such an observation confirms insights on the structural gradients of the structure within thin films[56], where crystallization occurs at different rates within varying thicknesses of the film by solvent evaporation and interdiffusion[57] and resultant colloidal assembly[58]. Moreover, the Bragg peaks of the solvent-complex lose intensity from the off-normal orientations faster, and the remaining intensities of the Bragg peaks from the solvent-complex display orientations dominantly normal to the film substrate (Figure S2). With continued annealing, the perovskite film crystal orientation becomes increasingly mosaic as seen from the increasing widths of the related Bragg peaks, and the homogenized distribution of the preferential orientation spread of the perovskite Bragg peaks (Figure S2b, c). During the gradual removal of the solvent complex, the intensity of the perovskite peak (Fig. 3a) shows distinct fluctuations, first diminishing ($t = 121$–130 s), then increasing ($t = 131$–136 s), and then decreasing again ($t = 137$–144 s). These fluctuations are attributed to dissolving and recrystallizing previously formed perovskite crystals as the solvent molecules from deeper interfaces are removed by solvent mass transfer and crystalline rearrangement through the already crystallized film thickness. The peak width of the perovskite crystals during this stage ($t = 121$–144 s) increases while the lattice parameters reflect a slight decrease (Fig. 3b), confirming the presence of processes that engender increased structural disorder within the thin film. Any discernible signs from the solvent-complex phase disappear as the film is fully converted into the dry crystalline phase (phase IV, $t = 144$ s).

Upon reaching 100 °C in phase III, a second bright PL response emerges between 650 and 740 nm (1.68–1.9 eV), with its center around 1.72 eV and a FWHM of 190 meV (Fig. 3d) observed at $t = 104$ s (Figure S1b). This signature is attributed to the

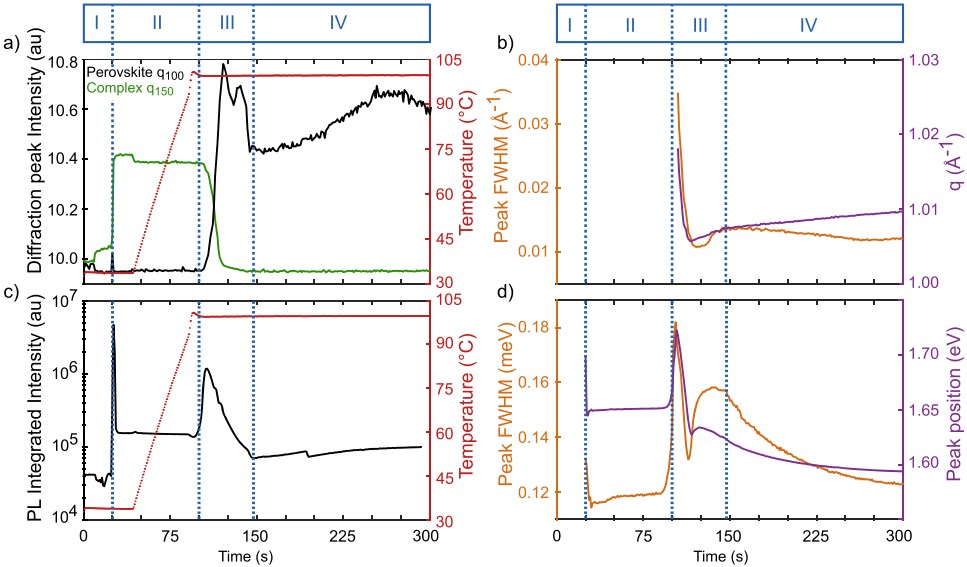

**Fig. 3 Temporal evolution of characteristic parameters. a** The temporal evolution of radially integrated GIWAXS data of MAPI (black) and MAPI·DMSO solvent-complex (green) as well as temperature (red) and (**b**) perovskite lattice parameters peak full width at half maximum (FWHM, orange) and q position (purple) during the crystallization process of perovskite solvent complex at 100 °C. **c** Integrated PL emission intensity (black) during nucleation, annealing, and dissolution-recrystallization processes of thin film as well as temperature (red) and (**d**) Evolution of PL peak parameters peak FWHM (orange) and position (purple) during crystallization experiment. The four phases are indicated with vertical dotted lines.

co-existence of disparate nanocrystallite sizes with high radiative efficiency. The non-Gaussian peak shape comes from a superposition of luminescence signals with disparate intensity contributions[47,59]. Akin to the processes occurring during the moments proceeding antisolvent dispensing ($t > 25$ s), the PL signal intensity decreases and redshifts, indicating growth of the nanocrystallites. The underlying growth kinetics are slower, while the redshift of the peak maximum is more significant, reaching 1.72 eV at $t = 103$ s. In parallel, the peak width decreases from 180 meV ($t = 103$ s) to 130 meV ($t = 114$ s) (Fig. 3d). After the peak width attains a local minimum ($t = 114$ s), a subsequent re-broadening to 158 meV ($t = 135$ s) is observed, correlated to a small peak position shift towards higher energies. These trends reflect the dissolution and creation of small clusters with higher bandgap, as the solvent from the deeper parts are removed, confirming the trends in the diffraction data.

**Phase IV**. The final phase IV ($t > 144$ s) represents exclusive diffraction signals from cubic MAPI at $q = 1.00$, $1.42$, $1.74$, $2.00$, $2.24$, $2.46$, $2.84$, $3.01$, and $3.48$ Å$^{-1}$. In this phase, the intensity of the perovskite peak increases (Fig. 3a) up to 260 s, while the peak width narrows (Fig. 3b), suggesting enhanced crystallinity and reduced lattice strain on longer annealing. The Bragg peak intensity distribution suggests a similar slight preferred orientation (Fig. S2c) of the crystals of the thin film normal to the plane of the substrate, as is observed within the transient structure (Fig. 2b) when the antisolvent is dispensed. Beyond 260 s, the peak intensity of the perovskite decreases slightly (Fig. 3a) as the peak width saw a slight increase, which might be indicative of the onset of beam damage. No PbI$_2$ is isolated although the perovskite peak broadening signifies increased disorder in crystals.

In phase IV, the gradual increase in the overall PL intensities (Fig. 3c) is accompanied by an FWHM narrowing and shift in peak position towards 780 nm (1.60 eV), representative of luminescence from a stabilized bulk MAPI (Fig. 3d). PL data however did not indicate beam damage. Long-term annealing and cooling of a

sample reflect the presence of a PbI$_2$ phase ($q\sim0.9$ Å$^{-1}$) as well as of a tetragonal MAPI phase ($q\sim1.4$ Å$^{-1}$) (Fig. S3).

## Discussion

While there are several reports on phase transformations and identification[20,21,28,60], detailed insights into the phenomena occurring at critical synthesis and phase transition stages are desirable. Importantly, the kinetic processes occurring during nucleation and dissolution-recrystallization have not been revealed in detail so far, and are the focus of the present study (Fig. 4).

Within the context of crystallization of colloidal systems from solutions, the crystallization processes are known to proceed by fluctuating solvodynamics resulting in initial "low-amplitude", long-wavelength density fluctuations[61,62] through a large volume, followed by the actual crystallization event (Fig. S4c). These fluctuations may be reflected in our diffraction data right before the emergence of the weak Bragg reflections from the supersaturated phase when the scattering signal shows an intermediate transition from colloidal halos (Fig. S4a) at $q = 0.5$, $1.8$, and $3.0$ Å$^{-1}$ to a broad background signal (Fig. S4b) at the moment the antisolvent is dropped to initiate the crystallization process. Chemically, by virtue of the high miscibility of chlorobenzene with DMSO and DMF[9] and its poor solubility with the perovskite solid precursors, a phase separation process occurs as excess solvent molecules are displaced from the sample by the antisolvent stream, ensuing a marked increase in the concentration of the solute species within the system creating the conditions for a phase transition process to transpire. This phase transformation process, which marks the phase boundary between the fluid colloidal precursor to the gel intermediate state, can proceed by one of two routes, namely LaMer nucleation[63–65] or kinetically arrested spinodal decomposition[66]. Both processes signify pathways of segregation and evolution of a new phase[67] from a melt, where nucleation-driven phase transitions have associated activation energy for the creation of metastable nuclei, which coalesce to form the solvent-complex intermediate. A kinetically arrested spinodal decomposition process leads to the

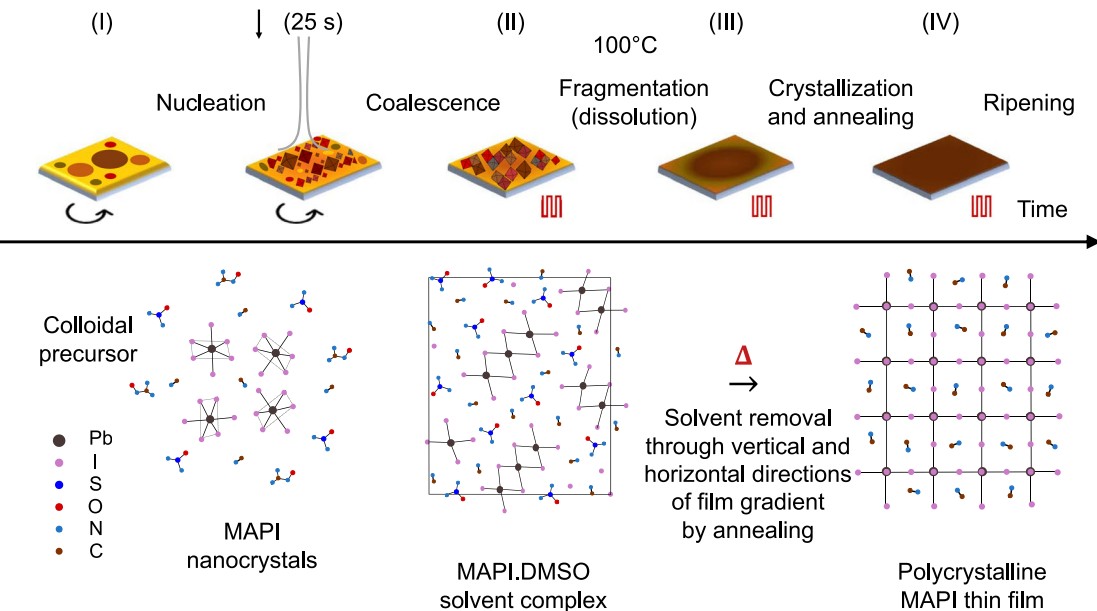

**Fig. 4 Four phases of film formation during spin coating.** Phase I: The thin film consists of colloidal precursors of photoinactive solid precursors (PbI$_2$ and CH$_3$NH$_3$I) and liquid solvents (CH$_3$)$_2$N-HCO and (CH$_3$)$_2$SO. Nucleation of MAPI nanocrystals and conversion to phase II where the nanoparticles trimerize into thermodynamically stable orthorhombic solvent-complex of Pb$_3$I$_8$.2(CH$_3$)$_2$SO.2CH$_3$NH$_3$ with remainders of stable perovskite phase formed during nucleation. Phase III: Thermal decomposition of solvent-complex leading to the removal of plumboiodide coordinated solvent molecules at 100 °C from the thin film and the eventual dissolution-recrystallization of CH$_3$NH$_3$PbI$_3$ from the film thickness. Phase IV: Perovskite crystallization complete and thermal ripening process of the thin film.

spontaneous formation of the said intermediate without the creation of metastable nuclei due to fluctuations in the energetics of the precursor. The impact of varying the antisolvent drop time during spin-coating strongly affects the material conversion pathways and resulting morphologies. Previous studies have extensively studied the impact of drop times and formulated and explained the concept of antisolvent drop time windows in separate studies[19,28,68]. In the absence of an antisolvent, materials crystallizing out of a fluid precursor nucleate and crystallize in a broad time window, whereas the application of an orthogonal solvent is a physical route to temporally regulate supersaturation and initiate growth during material crystallization/formation. Accordingly, there are distinct differences in precursor chemistry and methods of physicochemical conversion between a one-step[9] and two-step[20,69] method which are typically discussed in the literature. The one-step method combines the organic (CH$_3$NH$_3$I) and inorganic precursors (PbI$_2$) in a solvent system (4:1 v:v DMF:DMSO) to afford a single colloidal precursor. This precursor is spin-coated into a thin film, followed by an application of an antisolvent and subsequent thermal annealing to result in a CH$_3$NH$_3$PbI$_3$ film. The two-step method proceeds via the separated treatment of the inorganic (PbI$_2$) precursor solution-processed into a thin film with suitable organic solvents (GBL, DMF, DMSO), immersed into an organic precursor (CH$_3$NH$_3$I/isopropanol) to convert to CH$_3$NH$_3$PbI$_3$ via thermal annealing driven interdiffusion method. The removal of the bulk solvent molecules is expected to lead to a drastic reduction in the excluded free volume available for the solute molecules and cause a pinned gel-like structure[21,66] with decreased diffusion abilities. The stochastic nature[61,70] of nucleation of the perovskite precursor colloid[37,71], where a broad distribution of nuclei sizes are expected at supersaturation, is reflected in the evolution of the PL FWHM. Whereas antisolvent-driven nucleation is expected to lead to a homogenous nucleation event, experiments suggest that the actual nature of nucleation depends on the time window employed for antisolvent dispensing[19]. In our experiments, for the employed experimental conditions, an event resulting in a broad distribution

of nanoparticles is marked, as witnessed by the broad photoluminescence signal. Such a signal implies a broad distribution in the density of states of luminescing species, which corroborate a distribution in the nuclei sizes formed during nucleation. We note that this is the first observation of the pre-nucleation density fluctuations, (Fig. S4b) and nucleation (Fig. 2b) within crystallizing perovskite systems by combined diffraction and photoluminescence data. The nuclei are notably isostructural with the perovskite (MAPI) phase but owing to the large destabilizing surface and interfacial energies associated, the diffraction data of the nucleated phase is only briefly observed (Fig. 2b) before converting to the solvent-complex phase (Fig. 2c). The nucleating species likely have high surface charge, and cluster-aggregation of individual monomers (PbI$_6$$^{4-}$) to the trimerized orthogonal solvent complex (Pb$_3$I$_8$$^{2-}$) proceeds by means of increased entropy[72] (and reduced free energy) on the release of solvating molecules during the post-nucleation, early-growth stages. The DMSO molecules coordinate the trimerized aggregates, while the organic methylammonium (CH$_3$NH$_3$$^+$) ions are known to characteristically direct the self-assembly[73,74] of the structures resulting in the denser solvent-complex. The PL intensity is found to be influenced by two major factors: the nucleation density (total amount of perovskite material responding to the photoexcitation) and the extent of radiative recombination (depending on crystal quality and defect density[26,53]). The data supports the reduction in the polydispersity soon after the instance of nucleation by the growth in average particle size, as reflected within the narrowed shape of the PL peak and its redshift from 730–750 nm. The PL intensity during antisolvent-induced nucleation increases ($t = 25$–27 s) as shown in Fig. 3c, in agreement with previous reports[53] and is ascribed to an increasing number of nanocrystallites[50,53]. Subsequently, the intensity is quenched (30×) due to cluster coalescence; also reflected within the PL redshift. The coalesced clusters have reduced emission[75] as compared to the nucleated nanoparticles because with size increase, structures have a higher probability for non-radiative recombinations at defect sites and grain boundaries. A combination

of increased crystallite size and thermal quenching leads to an increase in the extent of non-radiative recombination and a resultant reduction in the luminescence intensity from the remaining stable MAPI crystals formed during nucleation [26,50].

The thermodynamic stability of the DMSO-complexated phase requires thermal annealing to drive the removal of the DMSO molecules. The reduced free solvent content within the film is associated with the limited diffusion ability of the media, until the energy barrier is overcome by heating the sample. Throughout the annealing process, the nature of the diffraction and photoluminescence signals remain largely unchanged, until 100 °C is reached and the disintegration process of the solvent complex is initiated, marking a subsequent second-order phase transition. During thermal disassociation of the MAPI·DMSO solvent complex, the diffraction intensities from the solvent complex diminish as the diffraction intensity of the perovskite phase gains in intensity (Fig. 3a). While the diffraction data reveals the two coexistent phases as they evolve, marking the second order of the phase transition, the PL data yields mechanistic insights underlying the transition. The disassociation of the solvent complex leads to the removal of DMSO by evaporation, leaving behind fragmented building blocks of the crystallizing MAPI phase. This process is deemed responsible for the increased polydispersity of the molecular species, reflected in the broadened and markedly blue-shifted PL signal during phase III. Owing to the increased surface energies of the disassociated particles, the fragments thermally diffuse to form longer chains to reduce the total free energy of the system. This process of the concatenation of plumboiodide fragments and their assembly within crystalline cubic perovskite lattice are reflected by shifts in the PL signal. The PL signal after its previous broadening and blue shift, undergoes rapid narrowing and redshifts as solvent molecules are rapidly removed from the thin film leaving MAPI crystals behind. It has been found that owing to differential solvent evaporation rates from the film-air interface and the deeper entrenched solvent moieties, the film develops a vertical gradient of solvent distribution resulting in a crystallizing front leading from the film-air interface into the film-substrate interface, creating a crust of crystallized perovskite at the surface [76].

The processes occurring during annealing and drying can be distinguished into distinct drying stages, with considerations of heat and mass transfer phenomena[77]. After the initiation of the annealing step in phase II, the temperature of the substrate increases linearly over time and by a heat-transfer process, the temperature of the thin film increases. In contrast, during phase III, the temperature is kept constant and the thermal disintegration of the solvent complex allows significant evaporation of the solvent to the air interface. The solvent removal falls within the "fast" regime[78] and is strongly affected by the coupled heat and mass transfer between the drying film interface and air. The process of solvent removal proceeds by diffusion and evaporation and therefore includes advective mass transfer through the bulk. The removal of the solvent molecules underneath the film-air interface is responsible for the secondary processes beyond 120 s, where both the PL and the diffraction peaks show fluctuations in intensity due to subsequent redissolution and crystallization within different depths and interfaces. Phase IV of the process of crystallization is marked by the full conversion of the solvent complex into the MAPI structure, with all structural and optoelectronic signatures of the complex disappearing, signifying the completion of the crystallization process.

In conclusion, the combination of in situ photoluminescence and grazing-incidence wide-angle X-ray scattering is used to follow in real-time the colloidal processing of perovskite thin films during spin coating. Advanced multimodal experimental observation of the structure and optoelectronic properties of the luminescent, metastable nucleated $CH_3NH_3PbI_3$ nanoparticles during processing is presented. These building blocks are tracked in real-time as they are transformed into the orthogonal solvent-complex $Pb_3I_8 \cdot 2(CH_3)_2SO \cdot 2CH_3NH_3$. During annealing, the solvent complex disintegrates, and a solvent gradient is established through the thin film leading to a crystallization-redissolution-recrystallization and rearrangement process throughout the film thickness. The final phase of $CH_3NH_3PbI_3$ is stabilized in the cubic symmetry and exhibits the expected structural and optoelectronic characteristics.

## Methods

**Multimodal experiment.** The experiment was carried out at the 12.3.2 micro-diffraction beamline at the Advanced Light Source in a custom-made analytical chamber, allowing for processing of the thin film and simultaneous multimodal measurements. The TiO₂ covered plasma cleaned glass substrate was placed onto the integrated spin coating puck-heater and held in place by a heat transfer paste. A liquid precursor of 1 M PbI₂ and CH₃NH₃I in a solvent mixture of 4:1 DMF:DMSO was pipetted and deposited onto the surface of the substrate and the chamber was sealed off from the external environment by being held under a nitrogen flow. The experiment was conducted by spin coating the precursor at two spin coating speeds, a first 10 s of slow rotation at 100 rpm to ensure the uniform spread of the precursor onto the substrate and a second 30 s of fast rotation at 3000 rpm in order to fabricate a thin film. 15 s into the second spin coating step ($t = 25$ s), a stream of chlorobenzene was dispensed through a pre-programmed syringe pump. At the end of the spin coating protocol, a heating protocol was remotely initiated, in two stages. In the first stage ($t = 45$–90 s), the temperature was increased linearly at the rate of 1 °C/ s until it reached and stabilized at 100 °C. Thereafter, the temperature was maintained at 100 °C until the end of the experiment ($t = 90$–300 s). The incident angle of the incoming X-ray beam was 1° with a beam energy of 10 keV. The sample detector distance (SDD) was ~155 mm and the detector was positioned at an angle of 39° from the sample plane.

**In situ measurements.** GIWAXS data were recorded every second on a 2D Pilatus 1 M detector (Dectris Ltd.). Photoluminescence excitation was achieved through a 532 nm Thorlabs diode-pumped solid-state laser with a power density of 40 mW/cm². The resultant photoluminescence signal was collected by a lens and focused on an optical fiber guiding it to a grating OceanOptics QE Pro spectrometer for detection. The temperature of the heating puck was recorded by a pre-calibrated Raytek MI3 pyrometer, which regulated the annealing temperature and protocol through a pre-programmed PID loop.

## Data availability

The data are available from the corresponding authors upon reasonable request.

## Code availability

The code used for data management and analysis are available from the corresponding author upon reasonable request.

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

## Acknowledgements

This work was supported by funding from TUM.solar in the context of the Bavarian Collaborative Research Project Solar Technologies Go Hybrid (SolTech) and by the Deutsche Forschungsgemeinschaft (DFG, German Research Foundation) under Germany´s Excellence Strategy – EXC 2089/1 – 390776260 (e-conversion). S.P. acknowledges support from the TUM International Graduate School of Science and Engineering (IGSSE) via the GreenTech Initiative Interface Science for Photovoltaics (ISPV) of the EuroTech Universities, the Bavaria California Technology Center (BaCaTeC), and the Centre for Nanoscience (CeNS). Special thanks are extended to the Nanosystems Initiative Munich (NIM) and the generous ALS Doctoral Fellowship in Residence for funding and administrative support that made this work possible. H. Goudey and the ALS workshop are acknowledged for their support with the development of the instrument. Dr. A. Sharma and L. P. Kreuzer helped with their critical insights. This research used resources of the Advanced Light Source, a U.S. DOE Office of Science User Facility under contract no. DE-AC02-05CH11231. This manuscript was prepared with support from the Laboratory Directed Research and Development (LDRD) program of Lawrence Berkeley National Laboratory under U.S. Department of Energy Contract DE-AC02-05CH11231(T.-B.S. and C.M.S.-F.). This material is based upon work performed by the Joint Center for Artificial Photosynthesis, a DOE Energy Innovation Hub, supported through the Office of Science of the U.S. Department of Energy under Award DESC0004993 (F.B.). C.M.S.-F. acknowledges the Molecular Foundry supported by the Office of Science, Office of Basic Energy Sciences, of the U.S. Department of Energy under Contract No. DE-AC02-05CH11231.

## Author contributions

C.M.S.-F., C.V.S., N.T., S.P. and J.L.S. conceived, designed, built, and deployed the instrumentation. S.P., N.S.B., Z.H., Z.Y., T.L., T.-B.S., J.L.S., C.V.S., N.T. and C.M.S.-F. conducted the experiment. C.V.S. and N.T. provided beamtime supervision and support. S.P., Z.Y., N.S.B. and F.B. developed methods for data analysis and representation. S.P., F.B., C.M.S.-F. and P.M.B. wrote the manuscript. C.M.S.-F. and P.M.B. supervised the project. All authors discussed and approved the final paper.

## Funding

## Competing interests

The authors declare no competing interests.
