## [Peer Review File · Nature Communications]

Out-of-equilibrium processes in crystallization of organic-inorganic perovskites during spin coatingREVIEWER COMMENTS

Reviewer #1 (Remarks to the Author):

The manuscript describes GIWAXS and PL in-situ measurements during the growth of MAPI thin films using an anti-solvent-based nucleation process and analyses the evolution and transformation of phases during the full process-duration from spin-coating to the crystallization of the final MAPI film. The time-resolution and signal quality of the experiments is very high and the analysis of the phenomena occurring during the solution-assisted growth of a MAPI film is very detailed and thoughtful. I think this manuscript is suited for Nature Communications, if improvements are made.

(1) Reproducibility of results. The manuscript shows a detailed analysis and description of a single process. It is known that reproducibility is a problem in (any) thin film processing, in particular also for solution processing.

(2) Lack of variation of parameters. There have been a number of studies (also cited by the authors) that the timing of the antisolvent introduction plays an important role for the film/device quality. Did they investigate this? Why was the specific antisolvent drop-time shown in the manuscript chosen.

(3) Discussion of liquid process with antisolvent and without antisolvent treatment. There are a number of reports of in-situ GIWAXS experiments of MAPI solution processes in the literature. The authors should explicitly discuss how the antisolvent route is different from the "standard" recipe.

(4) The manuscript is very well written but at some places, the authors do not distinguish clearly between what is a hypothesis and what is a (proven) fact, e.g. "Owing to differential solvent evaporation rates from the film-air interface and the deeper entrenched solvent moieties, the film develops a vertical gradient of solvent distribution with a crystallizing front leading from the film-air interface into the film-substrate interface, creating a crust of crystallized perovskite at the surface."
- It would be more appropriated to write such paragraphs using "it has been found, it is assumed, ..."

(5) Some sentences are difficult to understand e.g., "thermally induced degradation of the solvent-complex as a function of film thickness position on differential rates of removal"
- As a function of film thickness position?
- Differential rates of removal?

(6) Overloaded statements: "we unite processing-structure-function relationships within decisive experiments and unveil previously empirically inaccessible mechanistic insights of complex colloidal crystallization."
- what is the function in the processing-structure-function relationship?
- What do the authors mean by "decisive experiments" – to my knowledge they perform only 1 experiment.
- "Unveil mechanistic insights" also seems an overstatement, since the authors infer or hypothesize mechanistic processes.

(7) Ref 32 seems out of context

(8) "With continued annealing, the perovskite film crystal orientation becomes increasingly mosaic."
- What (or where) is the evidence for this statement?

(9) "Such an event results in LaMer nucleation^{61–63} or possibly a kinetically arrested spinodal decomposition"
- The authors should describe what they mean by "kinetically arrested spinodal decomposition"

(10) The stochastic of nucleation of the perovskite precursor colloid, where a broad distribution of

nuclei sizes are expected at supersaturation, is reflected in the evolution of the PL FWHM"

- It seems that the anti-solvent induced nucleation would lead to homogeneous nucleation; what is the reasoning for the authors assumption of a broad distribution of nuclei ?

(11) "We note that this is the first observation of the pre-nucleation density fluctuations and actual nucleation within crystallizing perovskite systems by combined diffraction and photoluminescence data."

- What is the evidence of "pre-nucleation density fluctuations" ? I could not find it.

(12) "Subsequently, the intensity is quenched (30x) due to cluster coalescence; also reflected within the PL red-shift."

-Why would cluster coalescence reduce the PL intensity ?

(13) Unclear process description: "Following 24 s of spin coating, an antisolvent stream is dynamically dispensed on the spinning film, resulting in rapid structural changes within the precursor phase"

-What does "dynamically dispensed" mean ? How does this compare to typical antisolvent recipes?

(14) The FWHM of the PL peak at 750 nm shows significant narrowing from 130 meV ($t = 26$ s) to 110 meV ($t = 29$ s) (Figure S1a), corroborating the

homogenization in size distribution. This occurs through an increase in the average sizes of structures formed by cluster collapse of the nanoparticles with high correlated surface and interfacial energies."

- What is this statement based on (the highly correlated surface and interface energies..) ?

(15) This signature is attributed to the co-existence of disparate nanocrystallite sizes with high radiative efficiency, while the non-Gaussian peak shape comes from a superposition of luminescence signals.

- What do the authors mean by "disparate nanocrystal sizes" and why would this be a nonGaussian ? size distributions usually lead to Gaussian shapes !

(16) "While there are several reports on phase transformations and identification, detailed insights into the phenomena occurring at critical synthesis and phase transition stages are desirable"

-please use references again here; And - your work seems specific to a very specific process, MAPI with antisolvent added at a very specific time and heating to 100C. Please make sure that this is clear and what means for possible generalizations (or need for further experiments)

(17) A convolution of increased crystallite size and thermal quenching lead to an increase in the extent of non-radiative recombination leading to a reduction in the luminescence intensity from the remaining stable MAPI crystals formed during nucleation.

- What do you mean by "convolution" here? This is a well-defined mathematical term

(18) Figure 1: He-Ne Laser usually has a wavelength of 632.8nm, not 690nm; how can this be explained ?

(19) Fig 1: Please label the phases/reflections

(20) Figure 3: It is very difficult to navigate through the figure and recognize what is actually shown since the y-axis are labeled with "intensity" . Please label the subfigures appropriately with the the measurement method

Reviewer #2 (Remarks to the Author):

The film formation of MAPbI₃ during spin-coating is studied in-situ with GIWAXS and photoluminescence. This is an interesting paper, that is mostly written clearly. I expect that it will be of some interest to the perovskite community, even though the results are not very surprising. I think the paper could be substantially improved with further experimental data.

Some questions need to be answered:

- 1) The line at 690 nm is said to originate from a HeNe laser. Please double check as a HeNe laser typically emits at 632.8 nm. Also in the experimental: "532 nm Thorlabs laser diode with a power density of 40 mW/cm²" The 532 nm laser is more likely a diode pumped solid state laser, that is frequency doubled. Please double check as well.
- 2) The strong PL after dropping the antisolvent is spectrally broad. The authors argue that this would be due to polydispersity of the MAPI crystallites. The authors claim quantum confinement. Can you give an estimate of the size of the particles from the x-ray data and correlate with the optical data? In addition, The authors say that in this stage the film consists of a crystalline solvent-complex, rather than the final perovskite material. Does that mean that the complex emits the light? This part needs better clarification.
- 3) I was missing the cooling of the final layer to RT. Are there any further changes? There is certainly a phase transition in between.
- 4) Please comment on the possible loss of MAI during the heating process. Decomposition of MAPI has been found already at lower temperatures. Thus, I would expect to find some PbI₂.
- 5) The authors claim that these kinds of in-situ studies are required to better understand film formation and to further improve on material quality. However, the paper does not show how the material quality could be improved, nor does the paper demonstrate the resulting material in any kind of application. In addition, MAPI is the most studied halide perovskite, but currently other compositions are more relevant for device applications. How can the insights of the present study be transferred to these more relevant materials?
- 6) The drying kinetics forming a crust at the perovskite-air interface may limit the thickness of a perovskite layer that can be processed by this procedure, because ultimately, the solvent from deep regions may no longer be able to leave the film. Can you comment on possible limitations? What is the influence of the final temperature and the ramp? You tested just one set of parameters.

Reply to comments of Reviewer #1 on manuscript NCOMMS-20-47684:

Comment: The manuscript describes GIWAXS and PL in-situ measurements during the growth of MAPI thin films using an anti-solvent-based nucleation process and analyses the evolution and transformation of phases during the full process-duration from spin-coating to the crystallization of the final MAPI film. The time-resolution and signal quality of the experiments is very high and the analysis of the phenomena occurring during the solution-assisted growth of a MAPI film is very detailed and thoughtful. I think this manuscript is suited for Nature Communications, if improvements are made.

Answer: The authors thank the reviewer for taking the time to read our manuscript and the positive and valuable comments, which helped us to further improve the manuscript. In particular, we are delighted that the reviewer judges the work suited for Nature Communications.

Comment: (1) Reproducibility of results. The manuscript shows a detailed analysis and description of a single process. It is known that reproducibility is a problem in (any) thin film processing, in particular also for solution processing.

Answer: The authors thank the reviewer for this helpful comment. We agree with the reviewer that solution based thin film processing is challenging as many dynamic parameters are involved in the process. Spin-coating is a non-equilibrium thin film preparation technique. The reproducibility of thin films fabricated is a well-known issue, which arises due to out-of-equilibrium processes, which are prevalent during film formation. Complementary and multimodal metrological techniques help to elucidate complex mechanisms responsible for reproducibility issues. The scope of the manuscript lies in extending the understanding of stochastic processes during complex physicochemical transformations, of which the crystallization of hybrid perovskites is a case study. The understanding of complex processes is more suitably discussed in probabilistic rather than deterministic terms. The multivariate nature and routes of crystal growth are discussed by Dove et al¹ depicting how starting from precursor molecules, materials may crystallize to their bulk form through multiple reaction pathways. It is possible for the simple MAPI composition to follow completely distinct growth pathways, which do not involve nucleated nanoparticles but rather, for instance proceeding to the direct formation of the solvent-complex under similar conditions by kinetically arrested spinodal decomposition (SI). Thus, it is important to fix and control the processing parameters, which we have done in the present study.

The following discussion is added to the manuscript. It reads:

“Spin-coating is a solution based processing method, which produces non-equilibrium thin films. The crystallization of colloidal precursors of hybrid perovskites has an inherent multivariate nature and is known to lead to reproducibility issues of film characteristics. Routes of crystal growth have been discussed, depicting how starting from generic precursor molecules, materials crystallize to their bulk form through multiple reaction pathways. Complementary and multimodal metrological techniques help to elucidate complex transformation mechanisms responsible for reproducibility issues. For instance, it is possible for the precursor to follow physicochemical growth pathways such as spinodal decomposition, which deviate from the reaction pathways involving traditionally nucleated species. Rather, for instance, chemical reactions may proceed through the formation of intermediates, which convert to their final structural form on further treatment. Conceptualizing an understanding of complex growth processes requires the fixing of processing parameters, which we have done in the present study.”

Comment: (2) Lack of variation of parameters. There have been a number of studies (also cited by the authors) that the timing of the antisolvent introduction plays an important role for the film/device quality. Did they investigate this? Why was the specific antisolvent drop-time shown in the manuscript chosen.

Answer: The authors thank the reviewer for this important comment. Many experimental parameters are known to affect crystallization processes, such as the solvent systems used², concentration of precursor utilized³, antisolvent drop time⁴ etc. Investigations on varying parameters such as the antisolvent drop time and its impact on the crystallization behavior are an important aspect that was the subject of extensive review^{2,4-6}. The specific anti-solvent drop time used in the present study was chosen in order to emulate the spin coating and annealing methods established in literature, that are known to lead to the formation of high quality thin films.⁷ For the judicious use of synchrotron beamtime, parameters of study were fixed instead of screening for parameters by in situ measurements. The antisolvent drop time can be varied in order to affect the final crystalline morphology. Studies on the antisolvent drop time were conducted by the authors previously to formulate the concept of different windows of antisolvent drop time, with the underlying differences caused by time drop time windows explicated in detail^{2,4,6}. Therefore, this parameter was no more in the focus of this in-situ study and we selected a fixed time.

We clarified this in the revised manuscript. It reads:

“The impact of varying the antisolvent drop time during spin-coating strongly affects the material conversion pathways and resulting morphologies. Previous studies have extensively studied the impact of drop times and formulated and explained the concept of antisolvent drop time windows in separate studies.^{2,4,6}”

Comment: (3) Discussion of liquid process with antisolvent and without antisolvent treatment. There are a number of reports of in-situ GIWAXS experiments of MAPI solution processes in the literature. The authors should explicitly discuss how the antisolvent route is different from the “standard” recipe.

Answer: The authors thank the reviewer for this comment. We agree that in-situ GIWAXS experiments of MAPI solution processes exist in the literature and have been included in the discussion.

In the absence of an antisolvent, materials crystallizing out of a fluid precursor nucleate and crystallize in a broad time window, whereas the application of an orthogonal solvent (antisolvent) is a physical route to temporally regulate supersaturation and initiate growth during material crystallization/formation. Accordingly, there are distinct differences between a one-step and two-step method. We add explanation about these differences based on the existing literature to the manuscript. It reads:

“In the absence of an antisolvent, materials crystallizing out of a fluid precursor nucleate and crystallize in a broad time window, whereas the application of an orthogonal solvent is a physical route to temporally regulate supersaturation and initiate growth during material crystallization/formation. Accordingly, there are distinct differences in precursor chemistry and methods of physicochemical conversion between a one-step⁷ and two-step⁸ method which are discussed in literature. The one-step method combines the organic (CH₃NH₃I) and inorganic precursors (PbI₂) in a solvent system (4:1v:v DMF:DMSO) to afford a single colloidal precursor. This precursor is spin coated into a thin film, followed by an application of an antisolvent and subsequent thermal annealing to result in a CH₃NH₃PbI₃ film. The two-step method proceeds via the separated treatment of the inorganic (PbI₂) precursor solution processed into a thin film with suitable organic solvents (GBL, DMF, DMSO), immersed into an organic precursor (CH₃NH₃I/isopropanol) to convert to CH₃NH₃PbI₃ via thermal annealing driven interdiffusion method.”

Comment: (4) The manuscript is very well written but at some places, the authors do not distinguish clearly between what is a hypothesis and what is a (proven) fact, e.g. “Owing to differential solvent evaporation rates from the film-air interface and the deeper entrenched solvent moieties, the film develops a vertical gradient of solvent distribution with a crystallizing front leading from the film-air interface into the film-substrate interface, creating a crust of crystallized perovskite at the surface.”
- It would be more appropriated to write such paragraphs using “it has been found, it is assumed, ...”

Answer: The authors appreciate the reviewers comments. We have followed the advice and performed the recommended changes throughout the entire manuscript. For details, please see highlighted positions in the resubmitted manuscript.

Comment: (5) Some sentences are difficult to understand e.g., “ thermally induced degradation of the solvent-complex as a function of film thickness position on differential rates of removal “
- As a function of film thickness position ?
- Differential rates of removal ?

Answer: We thank the reviewer for pointing this out and agree. The difficult sentences have been broken down and rewritten in order to communicate the message with greater clarity. Please see highlighted text in revised manuscript.

It reads:

“Complementary and multimodal metrological techniques help to elucidate complex transformation mechanisms responsible for reproducibility issues. For instance, it is possible for the precursor to follow physicochemical growth pathways such as spinodal decomposition, which deviate from the reaction pathways involving traditionally nucleated species. Rather, for instance, chemical reactions may proceed through the formation of intermediates, which convert to their final structural form on further treatment. Conceptualizing an understanding of complex growth processes requires the fixing of processing parameters, which we have done in the present study. In this work, we actuate the advantages of real-time investigation of the evolving structure and optoelectronic properties by combining grazing-incidence wide-angle X-ray scattering (GIWAXS) and photoluminescence (PL) spectroscopy while emulating the one-step antisolvent-assisted crystallization of a perovskite thin film. We chose to investigate methylammonium lead iodide ($\text{CH}_3\text{NH}_3\text{PbI}_3$, MAPI), whose structural intermediates and transformation kinetics have been extensively investigated. This was done as a manner of demonstrating that much remains to be learned about out-of-equilibrium assembly processes of exemplary model systems. We report on the characteristics of the metastable structure formed by the first order phase transition occurring during antisolvent induced nucleation from the colloidal precursor sol. Further, there is a partial transformation of the metastable nuclei by concatenation of nanostructures leading to the formation of a solvent-complex, with the solvent-complex being stable against thermally induced degradation up to 100 °C. Around 100 °C, a second order transformation process of the complex to MAPI is initiated, by evaporation of the solvent from the thin film. Physicochemical reaction gradients are found emergent and attributed to differential rates of removal of strongly coordinated solvent molecules by advection of the evaporating solvent molecules from the film thickness. This mechanism of solvent removal results in a process of dissolution-recrystallization to lead to the final MAPI thin film. Further annealing leads to a ripening process of crystals within the thin film. Evolutionary data signatures, physical concepts and characteristics within structure-function correlations learned from model system of MAPI are transferable to other chemical compositions of hybrid perovskites.”

Comment: (6) Overloaded statements: “we unite processing-structure-function relationships within decisive experiments and unveil previously empirically inaccessible mechanistic insights of complex colloidal crystallization.”

– what is the function in the processing-structure-function relationship?

- What do the authors mean by “decisive experiments” – to my knowledge they perform only 1 experiment.

- “Unveil mechanistic insights” also seems an overstatement, since the authors infer or hypothesize mechanistic processes.

Answer: We thank the reviewer for the constructive review. Following the suggestion, the statements have been simplified and overloaded terms have been removed. For the given example it reads: “Combining the complementary and reinforcing nature of information divulged by synchrotron radiation based X-ray and optical metrologies within controllable *in situ* processing environments, we unite processing-structure-function relationships. Herein the optoelectronic response is measured by means of photoluminescence spectroscopy tracking the varying functional optical response of changing structural entities traced by means of GIWAXS, during material processing by spin coating, antisolvent drop, and subsequent annealing to unveil previously empirically inaccessible mechanistic insights of complex colloidal crystallization.”

Comment: (7) Ref 32 seems out of context

Answer: We agree with the reviewer and removed Ref 32 in the revised version of the manuscript.

Comment: (8) “With continued annealing, the perovskite film crystal orientation becomes increasingly mosaic.”

- What (or where) is the evidence for this statement ?

Answer: The authors thank the reviewer for this helpful comment. With continued annealing, the intensity distribution of the Bragg peaks changes. The integrated intensity data showcasing this is now added to the SI (Figure S2c) alongside with a short addition of the text in the main manuscript. It reads:

“With continued annealing, the perovskite film crystal orientation becomes increasingly mosaic as seen from the increasing widths of the related Bragg peaks, and the homogenized distribution of the preferential orientation spread of the perovskite Bragg peaks (see Figure S2b,c).”

Comment: (9) “Such an event results in LaMer nucleation^{61–63} or possibly a kinetically arrested spinodal decomposition”

- The authors should describe what they mean by “kinetically arrested spinodal decomposition”

Answer: This is a very valid comment and we apologize for the unclear writing. The following addition to the text has been made for a growth process occurring via a kinetically arrested spinodal decomposition route. It reads:

“This phase transformation process which marks the phase boundary between the fluid colloidal precursor to the gel intermediate state can proceed by one of two routes, namely LaMer nucleation or kinetically arrested spinodal decomposition. Both processes signify pathways of segregation and evolution of a new phase⁹ from a melt, where nucleation driven phase transitions have an associated activation energy for the creation of metastable nuclei, which coalesce to form the solvent-complex intermediate. A kinetically arrested spinodal decomposition process leads to the spontaneous

formation of the said intermediate without the creation of metastable nuclei due to fluctuations in the energetics of the precursor.”

Comment: (10) The stochastic of nucleation of the perovskite precursor colloid, where a broad distribution of nuclei sizes are expected at supersaturation, is reflected in the evolution of the PL FWHM”

- It seems that the anti-solvent induced nucleation would lead to homogeneous nucleation; what is the reasoning for the authors assumption of a broad distribution of nuclei ?

Answer: The authors thank the reviewer for raising this comment. Correctly, it is expected that the addition of antisolvent would lead to a homogeneous nucleation event. However, experiments suggest that the actual nature of nucleation depends on the time window employed for antisolvent dispensing. In our experiments, for the employed experimental conditions, an event resulting in a broad distribution of nanoparticles is marked, as witnessed by the initially broad and red shifting photoluminescence signal. Such a signal implies a distribution in the density of states of luminescing species due to quantum confinement, which signify a distribution in the nuclei sizes. We clarify this in the revised manuscript. It reads:

“Whereas antisolvent driven nucleation is expected to lead to a homogeneous nucleation event, experiments suggest that the actual nature of nucleation depends on the time window employed for antisolvent dispensing.⁴ In our experiments, for the employed experimental conditions, an event resulting in a broad distribution of nanoparticles is marked, as witnessed by the broad photoluminescence signal. Such a signal implies a broad distribution in the density of states of luminescing species, which signify a distribution in the nuclei sizes formed during nucleation.”

Comment: (11) “We note that this is the first observation of the pre-nucleation density fluctuations and actual nucleation within crystallizing perovskite systems by combined diffraction and photoluminescence data.”

- What is the evidence of “pre-nucleation density fluctuations” ? I could not find it.

Answer: The authors thank the reviewer for this helpful comment! Pre-nucleation density fluctuations are observed in the present study and pointed out in the SI. Long-wavelength, low density fluctuations are responsible for an initial liquid-liquid phase separation via a small increase in density of material in the liquid phase, over a large volume, that sets the path for the crystallization process to proceed further. They result in a crystallization of MAPbI₃ via spinodal decomposition rather than nucleation. Notably, these fluctuations are expected to stabilize the system and lead to crystallization in the absence of an activation energy associated nucleation process. These fluctuations are reflected within a change of the colloidal halos from the precursor to a broad background visible in diffraction, (Fig S4, 25s). We clarify this by adding the link to the SI. It reads:

“We note that this is the first observation of the pre-nucleation density fluctuations and actual nucleation within crystallizing perovskite systems by combined diffraction and photoluminescence data (Fig S4).”

Comment: (12) “Subsequently, the intensity is quenched (30x) due to cluster coalescence; also reflected within the PL red-shift.”

-Why would cluster coalescence reduce the PL intensity ?

Answer: We thank the reviewer for this question. Cluster coalescence leads to a growth in size of particles by a reduction in the number of individual luminescing emitting moieties (nanoparticles).

This process explains the reduction in the luminescence intensity and the red shift of the PL peak energy. We reword the text to clarify this. It reads:

“The coalesced clusters have a reduced emission¹⁰ as compared to the nucleated nanoparticles because with size increase, structures have a higher probability for non-radiative recombinations at defect sites and grain boundaries. A combination of increased crystallite size and thermal quenching lead to an increase in the extent of non-radiative recombination leading to a reduction in the luminescence intensity from the remaining stable MAPI crystals formed during nucleation.^{11,12}”

Comment: (13) Unclear process description: "Following 24 s of spin coating, an antisolvent stream is dynamically dispensed on the spinning film, resulting in rapid structural changes within the precursor phase"

-What does “dynamically dispensed” mean ? How does this compare to typical antisolvent recipes?

Answer: The authors thank the reviewer for this comment. Dynamic dispensing of the antisolvent refers to the application of the antisolvent while the spin coating process is under way, and the sample is in motion (dynamic state) rather than at rest (static state). This terminology is now clarified for better understanding in the manuscript. It reads:

“dispensed dynamically (the film is spinning, in motion)”

Comment: (14) The FWHM of the PL peak at 750 nm shows significant narrowing from 130 meV (t = 26 s) to 110 meV (t = 29 s) (Figure S1a), corroborating the homogenization in size distribution. This occurs through an increase in the average sizes of structures formed by cluster collapse of the nanoparticles with high correlated surface and interfacial energies.”

- What is this statement based on (the highly correlated surface and interface energies..) ?

Answer: The authors thank the reviewer for this comment. The statement is based on the high surface to volume ratio and therefore high associated surface energies of nanoparticles. The presence of a broad distribution of nanoparticles suggests the presence of a broad size distribution of chemical species, and associated chemical interfaces and high interfacial energies. Nanoparticles have high surface energy, and a large number of nanoparticles have large associated interfacial energies, which are destabilizing in nature. The high surface and interfacial energies are strong driving forces for the cluster coalescence of the species. Coalescence results in a homogenization of size particles, which results in a narrowing of the PL spectra. We reworded the text to clarify this. It reads:

“The high surface and interfacial energies are possibly strong driving forces for the cluster coalescence of the nucleated species due to their high surface to volume ratio. Coalescence results in size homogenization of the particles which result in narrowing of the PL spectra.”

Comment: (15) This signature is attributed to the co-existence of disparate nanocrystallite sizes with high radiative efficiency, while the non-Gaussian peak shape comes from a superposition of luminescence signals.

- What do the authors mean by “disparate nanocrystal sizes” and why would this be a nonGaussian ? size distributions usually lead to Gaussian shapes !

Answer: The authors thank the reviewer for this comment. Superposition of a number of peaks from different sizes of nanoparticles, with disparate contributions lead to the emergence of non-Gaussian peak shapes. A well-defined Gaussian peak is expected from well-defined and equal contributions of luminescing species, which are not the case for out-of-equilibrium intermediate species. Moreover,

even for a Gaussian size distribution of nanoparticles, an overall Gaussian emission is not expected due to differential quantum yields per particle size. We reworded the text to clarify this. It reads: “This signature is attributed to the co-existence of disparate nanocrystallite sizes with high radiative efficiency. The non-Gaussian peak shape comes from a superposition of luminescence signals with disparate intensity contributions.”

Comment: (16) “While there are several reports on phase transformations and identification, detailed insights into the phenomena occurring at critical synthesis and phase transition stages are desirable” -please use references again here; And - your work seems specific to a very specific process, MAPI with antisolvent added at a very specific time and heating to 100C. Please make sure that this is clear and what means for possible generalizations (or need for further experiments)

Answer: The authors agree, this is a very good comment. We have added respective references. Moreover, we agree that we investigate a very specific process in terms of explicit experimental details and conditions, however, generalizations can be made as clarified in the answer to comment 4.

Comment: (17) A convolution of increased crystallite size and thermal quenching lead to an increase in the extent of non-radiative recombination leading to a reduction in the luminescence intensity from the remaining stable MAPI crystals formed during nucleation.

- What do you mean by “convolution” here? This is a well-defined mathematical term

Answer: The authors agree to the reviewers comment. The term combination is a better suited terminology and the word convolution has now been replaced.

Comment: (18) Figure 1: He-Ne Laser usually has a wavelength of 632.8nm, not 690nm; how can this be explained ?

Answer: The authors thank the reviewer for the helpful comment. The error has been corrected and the figure caption now reads:

“The narrow line emission at 690 nm is related to the diffuse reflection of the laser used in the position alignment system of the beamline.”

Comment: (19) Fig 1: Please label the phases/reflections

Answer: The authors thank the reviewer for the constructive suggestion. Figure 1 was changed accordingly.

Comment: (20) Figure 3: It is very difficult to navigate through the figure and recognize what is actually shown since the y-axis are labeled with “intensity” . Please label the subfigures appropriately with the the measurement method

Answer: The authors thank the reviewer for the good comment. Figure 3 was changed accordingly.

Reply to comments of Reviewer #2 on manuscript NCOMMS-20-47684:

Comment: The film formation of MAPbI₃ during spin-coating is studied in-situ with GIWAXS and photoluminescence. This is an interesting paper, that is mostly written clearly. I expect that it will be of some interest to the perovskite community, even though the results are not very surprising. I think the paper could be substantially improved with further experimental data.

Some questions need to be answered:

Answer: The authors thank the reviewer for taking the time to read our manuscript and the positive and valuable comments, which helped us to further improve the manuscript. In particular, we are happy that the reviewer judges the work of interest for the perovskite community.

Comment: 1) The line at 690 nm is said to originate from a HeNe laser. Please double check as a HeNe laser typically emits at 632.8 nm. Also in the experimental: "532 nm Thorlabs laser diode with a power density of 40 mW/cm²" The 532 nm laser is more likely a diode pumped solid state laser, that is frequency doubled. Please double check as well.

Answer: The authors thank the reviewer for the helpful comment. The error has been corrected and the figure caption now reads:

"The narrow line emission at 690 nm is related to the diffuse reflection of the laser used in the position alignment system of the beamline."

Moreover, the 532 nm laser is indeed a diode pumped solid state laser. We also corrected this mistake.

Comment: 2) The strong PL after dropping the antisolvent is spectrally broad. The authors argue that this would be due to polydispersity of the MAPI crystallites. The authors claim quantum confinement. Can you give an estimate of the size of the particles from the x-ray data and correlate with the optical data? In addition, The authors say that in this stage the film consists of a crystalline solvent-complex, rather than the final perovskite material. Does that mean that the complex emits the light? This part needs better clarification.

Answer: The authors thank the reviewer for the helpful comments and questions. Previous studies show that the size of the particles soon after antisolvent dropping are within 10 nm range¹³. After the polydisperse nucleation process, and cluster coalescence, the MAPbI₃ nuclei above a certain critical radius are expected to remain stable against cluster coalescence. These MAPbI₃ crystals, not bound into the solvent complex phase are hypothesized to contribute to the remaining luminescence intensity in phase II. The solvent complex does not luminesce. We clarify this in the revised manuscript. It reads:

"After the polydisperse nucleation process, the MAPI nuclei above a critical radius are expected to remain stable against cluster coalescence. These MAPI crystals, which are not bound into the solvent complex phase, are hypothesized to contribute to the remaining luminescence intensity in phase II (the solvent complex does not luminesce)."

Comment: 3) I was missing the cooling of the final layer to RT. Are there any further changes? There is certainly a phase transition in between.

Answer: The authors thank the reviewer for the expert comment. Indeed, on cooling the sample, structural changes were observed in the tetragonal phase. Respective data on cooling were added to the SI (Figure S3) for the case of crystallization through spinodal decomposition.

Comment: 4) Please comment on the possible loss of MAI during the heating process. Decomposition of MAPI has been found already at lower temperatures. Thus, I would expect to find some PbI₂.

Answer: The authors thank the reviewer for the expert comment. The annealing processes would lead to the loss of MAI leading to the creation of PbI₂ over time. Data on the decomposition of MAPbI₃ and the creation of PbI₂ was added to the revised SI (Figure S3).

Comment: 5) The authors claim that these kinds of in-situ studies are required to better understand film formation and to further improve on material quality. However, the paper does not show how the material quality could be improved, nor does the paper demonstrate the resulting material in any kind of application. In addition, MAPI is the most studied halide perovskite, but currently other compositions are more relevant for device applications. How can the insights of the present study be transferred to these more relevant materials?

Answer: The authors thank the reviewer for the critical comment. Indeed, MAPI is the most studied halide perovskite, which is the reason why we have chosen this relative simple system for the initial in situ study. Our claim is not that we can bring MAPI solar cells to outperform other perovskite compositions, which have demonstrated superior device efficiencies. We argue that we provide a better understanding of the very complex film formation process. Such type of fundamental understanding is highly valuable for designing preparation protocols ahead of simple trial and error approaches. Target of such design criteria is the material quality. Thus, we do not see a need for demonstrating an application of MAPI films, as this is well known from the literature anyhow. The transfer of the reported findings to other perovskite systems, which to the reviewer are of more relevance is in the abstract understanding of the different pathways towards crystallite formation and morphology control. We highlight important transferable high level concepts in the revised manuscript.

“We chose to investigate methylammonium lead iodide (CH₃NH₃PbI₃, MAPI), whose structural intermediates and transformation kinetics have been extensively investigated. This was done as a manner of demonstrating that much remains to be learned about out-of-equilibrium assembly processes of exemplary model systems. We report on the characteristics of the metastable structure formed by the first order phase transition occurring during antisolvent induced nucleation from the colloidal precursor sol. Further, there is a partial transformation of the metastable nuclei by concatenation of nanostructures leading to the formation of a solvent-complex, with the solvent-complex being stable against thermally induced degradation up to 100 °C. Around 100 °C, a second order transformation process of the complex to MAPI is initiated by evaporation of the solvent from the thin film. Physicochemical reaction gradients emerged and are attributed to differential rates of removal of strongly coordinated solvent molecules by advection of the evaporating solvent molecules from the film thickness. This mechanism of solvent removal results in a process of dissolution-recrystallization to lead to the final MAPI thin film. Further annealing leads to a ripening process of crystals within the thin film. Evolutionary data signatures, physical concepts and characteristics within structure-function correlations learned from model system of MAPI are transferable to other chemical compositions of hybrid perovskites.”

Comment: 6) The drying kinetics forming a crust at the perovskite-air interface may limit the thickness of a perovskite layer that can be processed by this procedure, because ultimately, the solvent from deep regions may no longer be able to leave the film. Can you comment on possible limitations? What is the influence of the final temperature and the ramp? You tested just one set of parameters.

Answer: The authors thank the reviewer for this comment. The differential rates of crystallization process at different interfaces is expected in a broad range of experimental conditions. This is because crystallization processes necessarily proceed via different interfaces. For thin films, there are two relevant interfaces where crystallization occurs, an air-film interface and a film-substrate interface. Owing to the geometrical makeup of the interfaces, physicochemical transformations occur first at the air-film interface, with the film-substrate interface contained within. Reactions at the film-substrate interface are affected by the reactions occurring in the thickness of the film and the film-air interface, and therefore lag behind. Such vertical film gradients are commonplace and are expected to be observed in a broad range of experimental conditions. The authors have tested many different sets of parameters, which include beyond the temperature and temperature ramp also the antisolvent dripping time. The latter is actually an important parameter as outlined in earlier work of the authors.⁴ However, the present multimodal in-situ studies are focusing on one particular parameter set, thereby giving an example of the relevant film formation phases of MAPI films prepared. We clarify this in the revised manuscript. It reads:

“Conceptualizing an understanding of complex growth processes requires the fixing of processing parameters, which we have done in the present study.”

References

1. Yoreo, J. J. de *et al.* Crystallization by particle attachment in synthetic, biogenic, and geologic environments. *Science* **349**, aaa6760; 10.1126/science.aaa6760 (2015).
2. Zhao, P. *et al.* Antisolvent with an Ultrawide Processing Window for the One-Step Fabrication of Efficient and Large-Area Perovskite Solar Cells. *Advanced materials (Deerfield Beach, Fla.)* **30**, e1802763; 10.1002/adma.201802763 (2018).
3. Pratap, S., Keller, E. & Müller-Buschbaum, P. Emergence of lead halide perovskite colloidal dispersions through aggregation and fragmentation: insights from the nanoscale to the mesoscale. *Nanoscale* **11**, 3495–3499; 10.1039/c8nr09853k (2019).
4. Song, T.-B. *et al.* Dynamics of Antisolvent Processed Hybrid Metal Halide Perovskites Studied by In Situ Photoluminescence and Its Influence on Optoelectronic Properties. *ACS Appl. Energy Mater.* **3**, 2386–2393; 10.1021/acsaem.9b02052 (2020).
5. Gu, E. *et al.* Robot-Based High-Throughput Screening of Antisolvents for Lead Halide Perovskites. *Joule*; 10.1016/j.joule.2020.06.013 (2020).
6. Bruening, K. & Tassone, C. J. Antisolvent processing of lead halide perovskite thin films studied by in situ X-ray diffraction. *J. Mater. Chem. A* **6**, 18865–18870; 10.1039/C8TA06025H (2018).
7. Jeon, N. J. *et al.* Solvent engineering for high-performance inorganic-organic hybrid perovskite solar cells. *Nat. Mater.* **13**, 897–903; 10.1038/NMAT4014 (2014).
8. Xiao, Z. *et al.* Efficient, high yield perovskite photovoltaic devices grown by interdiffusion of solution-processed precursor stacking layers. *Energy Environ. Sci.* **7**, 2619–2623; 10.1039/C4EE01138D (2014).
9. Schmelzer, J. W. P., Abyzov, A. S. & Möller, J. Nucleation versus spinodal decomposition in phase formation processes in multicomponent solutions. *The Journal of Chemical Physics* **121**, 6900–6917; 10.1063/1.1786914 (2004).
10. van Dijken, A., Makkinje, J. & Meijerink, A. The influence of particle size on the luminescence quantum efficiency of nanocrystalline ZnO particles. *Journal of Luminescence* **92**, 323–328; 10.1016/S0022-2313(00)00262-3 (2001).

11. Wagner, L. *et al.* Distinguishing crystallization stages and their influence on quantum efficiency during perovskite solar cell formation in real-time. *Sci. Rep.* **7**, 14899; 10.1038/s41598-017-13855-6 (2017).
12. Suchan, K., Just, J., Becker, P., Unger, E. L. & Unold, T. Optical in situ monitoring during the synthesis of halide perovskite solar cells reveals formation kinetics and evolution of optoelectronic properties. *J. Mater. Chem. A* **8**, 10439–10449; 10.1039/D0TA01237H (2020).
13. Parrott, E. S. *et al.* Growth modes and quantum confinement in ultrathin vapour-deposited MAPbI₃ films. *Nanoscale* **11**, 14276–14284; 10.1039/c9nr04104d (2019).

REVIEWERS' COMMENTS

Reviewer #1 (Remarks to the Author):

The authors have addressed the questions and comments raised in the review. Therefore I can support publication in nature comm.

in the conclusion the authors now write "Premier multimodal experimental observation ...", which sounds more like advertising or public relations than a scientific conclusion. Please remove the word "premier" - you could write "advanced multimodal.."

Reviewer #2 (Remarks to the Author):

The authors have done their best to respond to the questions and concerns of the referees. I think the paper could be published as is.

Reply to comments from reviewer 1 on manuscript NCOMMS-20-47684A:

Comment: The authors have addressed the questions and comments raised in the review. Therefore I can support publication in nature comm.

in the conclusion the authors now write "Premier multimodal experimental observation", which sounds more like advertising or public relations than a scientific conclusion. Please remove the word "premier" - you could write "advanced multimodal.."

Answer: We thank the reviewer for the very positive feedback. We agree with the reviewer and have changed the text accordingly.

Reply to comments from reviewer 2 on manuscript NCOMMS-20-47684A:

Comment: The authors have done their best to respond to the questions and concerns of the referees. I think the paper could be published as is.

Answer: We thank the reviewer for the very positive feedback.